# High prevalence of intestinal schistosomiasis in school-age children in the villages adjacent to Lake Chamo in the southern Rift Valley of Ethiopia

Yenenesh Ayele, Teklu Wegayehu, Daniel Woldeyes, Fekadu Massebo◯*

Department of Biology, Arba Minch University, Arba Minch, Ethiopia

* fekadu.massebo@amu.edu.et, fekadu_massebo@yahoo.com

## Abstract

### Background

The prevalence of intestinal schistosomiasis remains a challenge despite government efforts to eliminate the disease. This study aims to assess the prevalence of intestinal schistosomiasis in school-age children living in the villages surrounding Lake Chamo in southern Rift Valley of Ethiopia.

### Methodology/Principal findings

A cross-sectional study was conducted from January to July 2023 in Shele Mela *Kebele* in Gamo Zone, South Ethiopia. Stool samples were collected from 597 school-age children. A singe Kato-Katz for helminths and formalin-ether concentration technique for protozoan parasites were used to process the samples. The intensity of *Schistosoma mansoni* infection among school-age children was determined by counting the number of eggs per gram of stool. Of the 597 children screened, 52.3% (95% CI: 48.4.4–56.4) were positive for *Schistosoma mansoni*. These findings showed that 55% of the infections were light, 30.3% were moderate, and 14.7% were heavy. The mean egg count of *S. mansoni* parasites was 182.1 eggs per gram. The prevalence of other intestinal parasites (*Hymenolopis nana*, *Ascaris lumbricoides*, Hookworm, *Taenia* species, *Giardia lamblia* and *Entamoeba histolytica*) was found to be 7.7% (46/597). The overall prevalence of *S. mansoni* co-infection with other intestinal parasites was 5.0% (30/597). Specifically, the co-infection rates were 1.5% for *A. lumbricoides*, 1.3% for *H. nana*, 1.0% for *Taenia* species, 0.2% for Hookworm, 0.2% for *E. histolytica*, and 0.2% for *G. lamblia*.

### Conclusions/Significances

The study showed a high rate of *S. mansoni* infection among school-age children. This calls for immediate action, such as school-based deworming, to protect these children from the

**Editor:** jong-Yil Chai, Seoul National University College of Medicine, REPUBLIC OF KOREA

**Data Availability Statement:** The authors confirm that all data underlying the findings are fully

available without restriction. All relevant data are within the manuscript.

**Funding:** The Norwegian Programme for Capacity Development in Higher Education and Research for Development (QZA-21/0162 to FM). The funders had no role in study design, data collection and analysis, decision to publish, or preparation of the manuscript.

**Competing interests:** The authors have declared that no competing interests exist.

disease and reduce the burden. Further research is needed to understand the transmission of the infection by the intermediate host.

## Author summary

Schistosomiasis is a neglected tropical parasitic disease that affects over 220 million people globally. Although there is a deworming mass drug administration (MDA) program for soil-transmitted helminths, the study area still needs to be integrated into the Praziquantel MDA. The study focused on school-age children to better understand the prevalence of the disease and recommend interventions. Hence, we assessed the prevalence and intensity of intestinal schistosomiasis among school-age children living in selected villages near Lake Chamo in the southern Rift Valley of Ethiopia. The results revealed that the prevalence of *Schistosoma mansoni* was 52.3%. The overall prevalence of other intestinal parasites (*Hymenolopis nana*, *Ascaris lumbricoides*, Hookworm, *Taenia* species, *Giardia lamblia*, and *Entamoeba histolytica*) was 7.7%. We documented a 5% *S. mansoni* co-infection with all other intestinal parasites including *A. lumbricoides*, *H. nana*, *Taenia* species, Hookworm, *E. histolytica*, and *G. lamblia*. This implies a high prevalence of intestinal parasites in school-age children, necessitating immediate action to alleviate this burden. Implementing school-based mass drug administration (MDA) for schistosomiasis is advisable in the study villages.

## Introduction

Schistosomiasis is a neglected tropical parasitic disease that affects more than 220 million people worldwide [1]. The tropical and sub-tropical regions are known to harbor the poorest individuals mostly affected by this disease [2]. Sub-Saharan African countries bear more than 90% of the schistosomiasis burden [3].

Seven species of *Schistosoma* parasites infect humans, namely *Schistosoma mansoni*, *Schistosoma haematobium*, *Schistosoma japonicum*, *Schistosoma mekongi*, *Schistosoma intercalatum*, *Schistosoma guineensis*, and *Schistosoma malayensis* [4]. *Schistosoma mansoni*, *S. haematobium* and *S. japonicum* are the most commonly found in human infections, with *S. mansoni* and *S. haematobium* being more prevalent in Africa [5]. Many parts of Africa have snail species that act as intermediate hosts for schistosomiasis. *Biomphalaria* species are intermediate hosts of intestinal schistosomiasis due to *Schistosoma mansoni*, while *Bulinus* species are intermediate hosts of urogenital schistosomiasis [6]. Schistosomiasis is a prevalent disease in Ethiopia, putting millions of people at risk of infection [7]. Two types of parasites, *Schistosoma mansoni* and *S. haematobium*, are commonly found in many areas in Ethiopia [8].

The primary strategy for combating schistosomiasis involves mass deworming at-risk individuals using praziquantel [1]. Ethiopia adheres to the WHO guidelines and implements mass drug administration (MDA) for school-age children as part of its deworming program whenever eligible for the MDA. While this approach has effectively reduced the disease burden in the community, it does not prevent reinfection if individuals come into contact with infected water sources [1]. Therefore, it is necessary to manage snails and maintain a hygienic environment to prevent water pollution from human waste.

Most people in villages near the southern Rift Valley lakes in Ethiopia rely on Lake Chamo and Lake Abaya, as well as the rivers that feed into these lakes, for irrigation, fishing, and other

household activities. This increased contact with water bodies puts people at a higher risk of *Schistosoma* infection [9]. Even though most people have access to water bodies, school-age children are more likely to make swimming a regular activity and, therefore, may be at a higher risk of infection than other groups. The study specifically targeted school-age children to better understand the prevalence of the disease and suggest possible interventions.

Apart from the soil-transmitted helminths deworming program, the study area has yet to be included in the praziquantel MDA due to insufficient evidence on the prevalence and intensity of schistosomiasis. The current study's findings support the implementation of essential preventive measures for those at higher risk of infection. This data could assist in efforts to prevent the disease at healthcare facilities and within various governmental and non-governmental organizations. Therefore, the current study aimed to determine the prevalence and intensity of intestinal schistosomiasis among school-age children living in selected villages near Lake Chamo in the southern Rift Valley lakes of Ethiopia.

## Materials and methods

### Ethics statement

The Arba Minch University Institutional Review Board (IRB/1254/2022) approved the study protocol. We also obtained informed written consent and assent from all participants and their parents/guardians. All positive cases were treated in accordance with the national treatment guidelines.

### Study area description

The study was conducted in Shele Mela *Kebele* (the smallest administrative unit), in the Arba Minch area district of South Ethiopia ([Fig 1]). *Kebele* consists of five sub-villages near Lake Chamo. Two health posts offer primary health care services in conjunction with the Kolla Shele Health Centre, located about 35 kilometers from Arba Minch town, the administrative Centre for the Gamo Zone.

Most people in the study area depend on agriculture as their primary source of income. They cultivate crops like bananas, avocados, mangoes, and maize and rely on irrigation from Lake Chamo, the Sille, and Sago rivers, which flow into the lake. It's important to note that

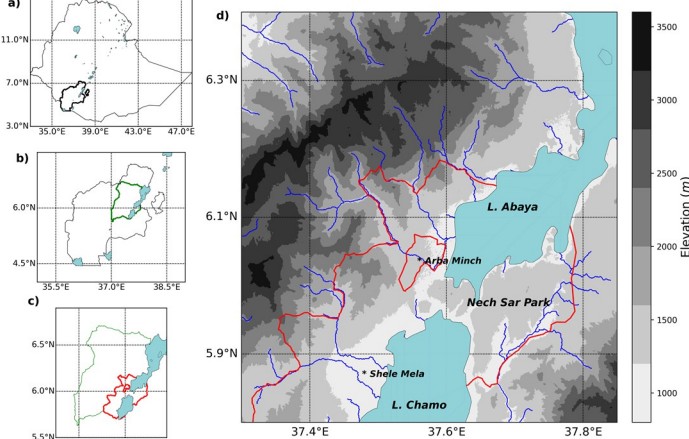

**Fig 1. Map of Shele Mela Kebele Arba Minch area district, Gamo zone, south Ethiopia.** U.S. Geological Survey (USGS): (http://www.usgs.gov), Roads, Water (River, lakes))–from DIVA-GIS (https://esri.maps.arcgis.com/apps/mapviewer/index.html).

intestinal schistosomiasis has been recorded in the Health Centre registration book and is a significant public health concern in the current study area. The presence of water bodies and frequent human contact with water, such as washing clothes, swimming, irrigating plantations, and fishing, could increase the risk of schistosomiasis. The study village was selected based on observing *Schistosoma* cases in the Health Centre, the availability of water bodies, and the human activities associated with those water bodies.

## Study design and period

A community-based cross-sectional study was conducted from January to July 2023 to determine the prevalence of intestinal schistosomiasis and other intestinal parasitic infections in school-age children.

## Study population, study participants and sampling technique

The study population was all school-age children (ages 5 to 15) in Shele Mela *Kebele*. We randomly selected households with school-age children in each sub-village to ensure that our samples were representative. The household lists were obtained from the registration book at the health post, and households with school-age children were extracted from the documents. After identifying the households from the lists, a statistician with no connection to the study conducted randomization. Trained data collectors visited the selected households, explained the study's purpose, and obtained informed consent from the parents or guardians of the children. The children were then given instructions on collecting and providing stool samples for analysis.

## Sample size determination

The single population proportion formula was used to calculate the sample size. This study comprised 597 school-age children. Because there had been no previous research in the area, the sample size was calculated using 50% of the prevalence.

Therefore, the sample was calculated as:

$$n = \frac{(Z\alpha/2)2 * Pq}{(d)2}$$

**Where, Z** = is table value of 5% level of significance (z = 1.96)
$\alpha$ = is level of significance ($\alpha$ = 0.05 = 5%),
d = is error tolerance (d = 0.04),
p = proportion of success (p = 0.5 = 50%)
q = proportion of failure (q = 1-p) = 0.5

## Inclusion and exclusion criteria

The study enrolled all school-aged children who lived in the selected households and voluntarily provided stool samples. Any child who was ill and receiving medication was excluded from the study. In addition, children whose parents or guardians did not provide consent were also excluded.

## Stool sample collection and examination

The children were taught how to collect stool samples without contaminating them. They were given applicator sticks, clean paper, and a labelled stool cup with details about the participant, such as age and sex. A portion of each stool sample was used to prepare a Kato-Katz thick

smear in the field. The remaining samples were placed in an icebox and transported to the College of Medicine and Health Sciences parasitological laboratory. They underwent parasite screening the same day, and any positive cases received immediate treatment.

### Kato-Katz technique

Stool samples were collected from children in selected households, and a smear was prepared according to the manufacturer's instructions for Kato-Katz thick smear preparation. An applicator stick was used to transfer a small portion of the stool sample onto a mesh-covered paper, which was then strained and screened with a spatula. The filtered sample was placed in the hole of a template at the center of a glass slide. After removing the template, the glass slide was covered with malachite green-soaked cellophane tape and pressed with another clean slide to spread the sample throughout the cellophane tape cover.

The slides were examined under an Olympus CX31 microscope with low-objective lenses (10 xs or 40 xs) within an hour of smear preparation. Positive results were confirmed with a medium power objective lens. The number of helminthes eggs present in the whole smear was counted and documented on the laboratory report form designed for this purpose.

### Concentration technique

To prepare the sample, it was mixed with 10% formal saline in a test tube and filtered through a 350–450 μm mesh. 7 ml of the filtrate was poured into a 15 ml centrifuge tube, and 3 ml of diethyl ether was added to the test tube. The mixture was carefully mixed and then centrifuged at 1500 rpm for 4 minutes, resulting in three layers. The top layer (diethyl ether-organic debris mixture) and the supernatant were removed, and a drop of sediment was taken from the remaining layers. This sediment was then placed on a glass slide, covered with a cover slide, and examined under a low-power microscope. This technique was used to check if parasites were missed by the Kato-Katz technique for treating cases, not for reporting results, as our primary focus was intestinal schistosomiasis.

### Quality assurance

The laboratory process began with a quality check of the chemicals and materials. Experienced laboratory technologists then applied the Kato-Katz and concentration techniques to diagnose.

### Data management and analysis

To summarize the data, descriptive statistics such as frequency and percentages were used. The Kato-Katz thick smear method counted the total number of parasite eggs in the stool sample. This count was then multiplied by 24 to determine the number of eggs per gram of stool. Based on this count, infections were classified as light, moderate, or heavy. The 95% confidence interval (CI) was presented along with the prevalence data.

## Results

### Prevalence of intestinal schistosomiasis infection

A total of 597 school-aged children participated in the study. Of these, 308 (51.6%) were female and 289 (48.4%) were male. The median age of the study participant was 8. Out of the 597 children who participated in the study, 312 of them were found to be infected with *S. mansoni*, which accounts for 52.3% (95% CI: 48.4.4–56.4) (Table 1).

**Table 1. Intestinal *Schistosoma* and other intestinal parasites co-infection prevalence among school-age children (n = 597) in Shelle Mela, south Ethiopia (January to July 2023).**

| Parasites species | Number of cases | Percentage (95% CI) |
|---|---|---|
| *S. mansoni* | 312 | 52.3 (48.2–56.3) |
| *H. nana* | 8 | 1.3 (0.6–2.6) |
| *A. lumbricoides* | 4 | 0.7 (0.2–1.7) |
| *G. lamblia* | 1 | 0.2 (0.004–0.9) |
| *Taenia* species | 3 | 0.5 (0.1–1.4) |
| *S. mansoni* + *A. lumbricoides* | 9 | 1.5 (0.7–2.8) |
| *S. mansoni* + *H. nana* | 8 | 1.3 (0.6–2.6) |
| *S. mansoni* + *Taenia* species | 6 | 1.0 (0.4–2.2) |
| *S. mansoni* + *Hookworm* | 1 | 0.2 (0.004–0.9) |
| *S. mansoni* + *A. lumbricoides* + *H. nana* | 1 | 0.2 (0.004–0.9) |
| *S. mansoni* + *A. lumbricoides* + *Taenia* species | 2 | 0.3 (0.04–1.2) |
| *S. mansoni* + *E. histolytica* | 1 | 0.2 (0.004–0.9) |
| *A. lumbricoides* + *E. histolytica* | 1 | 0.2 (0.004–0.9) |
| *S. mansoni* + *G. lamblia* | 1 | 0.2 (0.004–0.9) |
| Total positive | 358 | 60 (55.9–63.8) |

## Co-infection of *S. mansoni* and other intestinal parasites

The prevalence of co-infection of *S. mansoni* with other intestinal parasite was 5% (30/597; 95% CI: 3.4–7.1). There was co-infection of *S. mansoni* with all the parasites identified in this study. The study also documented the prevalence of triple infection, specifically *S. mansoni* with *A. lumbricoides* and *Taenia*, and *S. mansoni* with *A. lumbricoides* and *H. nana*. The prevalence of single infection of *H. nana* was 1.3% (8/597: 95% CI: 0.6–2.6), *A. lumbricoides* was 0.7% (4/597; 95% CI: 0.2–1.7), *Taenia* species was 0.5% (3/597; 95% CI: 0.1–1.4) and *G. lamblia* was 0.2% (1/597; 95% CI: 0.004–0.9) (Table 1).

## Prevalence of other intestinal parasites infection

Of 597 study participants, 48 (8%; 95% CI: 5.9–10.5) were positive for other intestinal parasites. The *H. nana* and *A. lumbricoides* were relatively common among helminths. Prevalence of infections due to *H. nana* was 2.8% (95% CI: 1.7–4.5) (17/597), and *A. lumbricoides* was 2.8% (95% CI: 1.7–4.5) (17/597) was relatively higher than that of the rest. Prevalence of Hookworm infection was found to be the lowest (0.2%; 95% CI: 0.004–0.9)). Among protozoans, *G. lamblia* and *E. histolytica* infections were rarely identified (Table 2).

**Table 2. Prevalence of other intestinal parasites co-infections among school-age children (n = 597) in Shele Mela, south Ethiopia (January to July 2023).**

| Parasite species identified | Number of positives | Prevalence |
|---|---|---|
| *H. nana* | 17 | 2.8 (1.7–4.5) |
| *A. lumbricoides* | 17 | 2.8 (1.7–4.5) |
| *Taenia* species | 9 | 1.5 (0.7–2.8) |
| Hookworm | 1 | 0.2 (0.004–0.9) |
| *E. histolytica* | 2 | 0.3 (0.04–1.2) |
| *G. lamblia* | 2 | 0.3 (0.04–1.2) |

### Parasite intensity

Among school-age children, the mean egg count of parasites was 182.1 eggs per gram. The intensity of *S. mansoni* infection ranged from light to heavy, with light (172/312: 55%; 95% CI: 49.4–60.7), moderate (95/312: 30.4%; 95% CI: 25.4–35.8), and heavy (45/312; 14.4%: 10.7–18.8). The mean number of eggs per gram of stool samples was 725.02 among heavily infected individuals, 195.8 in moderately infected ones, and 44.4 in lightly infected children. On the other hand, the parasite intensity of *H. nana*, *A. lumbricoides*, *Taenia* species, and Hookworm was moderate.

## Discussion

The study aimed to assess the prevalence and burden of *S. mansoni* infection among school-aged children in Shele Mela, located in south Ethiopia. The results indicate that 52.3% of school-aged children were infected with *S. mansoni*. The infection severity varied from mild to severe, with about 55% of the cases classified as mild, 30% as moderate, and 15% as severe, based on the WHO guidelines.

The study reveals a high infection rate of *S. mansoni* among school-age children, consistent with other studies in different parts of the country [10–12]. For instance, a 50% infection rate was found in Abbey and Didessa Valleys, Western Ethiopia [13]. Different prevalence rates have been reported across the country, with some areas showing higher prevalence than others [11,12,14]. Despite this variation, *Schistosoma* infection remains a significant public health concern in the country. The participants in the study have a habit of swimming and washing their clothes on river shores, increasing the risk of *S. mansoni* infection. Therefore, effective praziquantel deworming initiatives can lead to immediate health improvements, but their long-term success depends on a comprehensive approach. It's essential to involve the community and raise awareness about the significance of sanitation and hygiene practices, such as improved latrine usage. These practices can support deworming efforts and significantly increase the likelihood of success.

This study revealed that the overall prevalence rate of intestinal parasites, including co-infections and triple infections, was 8%. The identified soil-transmitted helminths in this area included *H. nana*, *A. lumbricoides*, protozoa such as *G. lamblia*, and *Taenia* species, as documented in several studies across different regions [8]. The prevalence rate of co-infection of *S. mansoni* with other intestinal parasites, including *Ascaris*, *H. nana*, *Taenia* species, Hookworm, *E. histolytica*, and *G. lamblia*, was 7.8%. Moreover, this study identified a triple infection of *S. mansoni* with *Ascaris*, *H. nana*, and *Taenia* species. Multiple co-infections of diseases may be linked to poor hygiene conditions [15], indicating a lack of access to clean water and unhygienic conditions and contact with contaminated water.

In this study, the severity of infection caused by *S. mansoni* was determined by measuring the number of eggs in a gram of stool. Light infections were observed in 55.0% of the cases, moderate infections in 30.3%, and heavy infections in 14.7%. These findings are consistent with a previous study conducted in different parts of Ethiopia [16]. The majority (85.5%) of positive cases had mild to moderate infections, possibly due to the initial number of infecting cercariae. Cercariae are found where the intermediate host (snails) is present and could potentially infect children in small numbers as they enter and exit the lake, typically involving brief water contact. *Schistosoma* infection results in premunition, where the initial infection triggers the body to prevent another similar parasite from entering, regardless of its burden [17]. There was no deworming program for *Schistosoma* infection in the study area, so most children may live with the infection chronically with low to moderate burdens.

This study has several limitations and strengths. The study did not investigate the identification of intermediate hosts. The study also used a cross-sectional design, which may lead to over- or underestimation of prevalence as it captures only the current situation and may need to reflect changes over time accurately. The study did not report the results of the concentration technique to compare with the Kato-Katz method. There might be better options for other intestinal parasites, although it is the right choice for abdominal schistosomiasis. Therefore, the results may not accurately reflect the prevalence of other intestinal parasitic infections. While the transmission-related factors are well-established and well-described in the study area description and introduction section, there was a missed opportunity for further exploration. This could have significantly benefited the current study. Despite this, the study's strengths, such as following standardized procedures conducted by experienced laboratory technicians, representative sampling, and minimized selection bias by including all school-age children in all selected households, still provide valid results that can aid the control program in reducing the disease burden.

## Conclusions

The study identified that more than half of school-aged children were infected with *S. mansoni*, with varying levels of infection severity. Therefore, a school-based deworming programme against *S. mansoni* is recommended to reduce the disease burden in children. Moreover, to ensure the sustainability of the deworming programme, it is important to raise community awareness about the significance of proper waste disposal and hygiene practices such as latrine usage.

## Acknowledgments

We acknowledge the parents/guardians for consenting to their children's participation in the study and the laboratory technologists for their support. We are grateful for Dr. Thomas Torora's contributions to mapping the study area.

## Author Contributions

**Conceptualization:** Yenenesh Ayele, Fekadu Massebo.

**Data curation:** Fekadu Massebo.

**Formal analysis:** Yenenesh Ayele.

**Funding acquisition:** Fekadu Massebo.

**Investigation:** Yenenesh Ayele.

**Methodology:** Yenenesh Ayele, Teklu Wegayehu, Daniel Woldeyes, Fekadu Massebo.

**Resources:** Fekadu Massebo.

**Supervision:** Teklu Wegayehu, Daniel Woldeyes, Fekadu Massebo.

**Writing – original draft:** Fekadu Massebo.

**Writing – review & editing:** Yenenesh Ayele, Teklu Wegayehu, Daniel Woldeyes, Fekadu Massebo.

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
