## [Decision Letter · Decision Letter 0]

19 Aug 2024

Dear Dr Massebo,

Thank you very much for submitting your manuscript "High prevalence of intestinal schistosomiasis in school-age children in the villages adjacent to Lake Chamo in southern Rift Valley of Ethiopia: an implication for schistosomiasis deworming?" for consideration at PLOS Neglected Tropical Diseases. As with all papers reviewed by the journal, your manuscript was reviewed by members of the editorial board and by several independent reviewers. In light of the reviews (below this email), we would like to invite the resubmission of a significantly-revised version that takes into account the reviewers' comments. 

We cannot make any decision about publication until we have seen the revised manuscript and your response to the reviewers' comments. Your revised manuscript is also likely to be sent to reviewers for further evaluation.

Sincerely,

Yaobi Zhang, M.D., Ph.D.

Guest Editor

Jong-Yil Chai

Section Editor

Reviewer's Responses to Questions

**Key Review Criteria Required for Acceptance?**

**Methods**

-Are the objectives of the study clearly articulated with a clear testable hypothesis stated?

-Is the study design appropriate to address the stated objectives?

-Is the population clearly described and appropriate for the hypothesis being tested?

-Is the sample size sufficient to ensure adequate power to address the hypothesis being tested?

-Were correct statistical analysis used to support conclusions?

-Are there concerns about ethical or regulatory requirements being met?

Reviewer #1: Yes, although the authors could describe a little more the parameters for the villages selected. They state households within villages randomly selected, which is fine, but not why certain villages surrounding the lake were selected. What was the criteria for deciding if the village was included? was there a maximum distance? Also, some more description of the study area itself, and why it hasn't been included previously in the PZQ program.

Reviewer #2: Yes.

Reviewer #3: -Are the objectives of the study clearly articulated with a clear testable hypothesis stated?

Neither the objectives nor study hypothesis were stated. 

-Is the study design appropriate to address the stated objectives?

Although a cross-sectional study design is suitable for assessing prevalence, it's challenging to evaluate alignment with the study's objectives without knowing them.

-Is the population clearly described and appropriate for the hypothesis being tested?

SAC 5-15 years was cited as study population, but no hypothesis was stated. 

-Is the sample size sufficient to ensure adequate power to address the hypothesis being tested?

Hard to tell. The estimated sample size was 597 SAC, the total population of the study area is unknown. Again, no hypothesis was cited. 

-Were correct statistical analysis used to support conclusions?

Inferential and descriptive statistics were used, but the conclusions were somehow vague.

-Are there concerns about ethical or regulatory requirements being met?

I would assume ethical concerns were addressed since the Arba Minch University Institutional Review Board approved the

study protocol and informed written consent and assent was obtained as needed. Also, positive cases were treated in accordance

with the national treatment guidelines.

**Results**

-Does the analysis presented match the analysis plan?

-Are the results clearly and completely presented?

-Are the figures (Tables, Images) of sufficient quality for clarity?

Reviewer #1: Yes, apart from the following: 1) the figure. Please describe what is being presented in the figure either in text or expand on the figure description.

2) It would be good to present the intensity as % prevalence of whole sample. ie ~28% of SAC in Shele Mela Kebele have light intensity infection X% moderate etc. This is because the WHO guidelines for elimination as a public health problem are tied to data presented in this manner

Reviewer #2: All prevalence estimates need to be accompanied by their 95% confidence intervals.

The results of the questionnaire also need to be presented.

Reviewer #3: -Does the analysis presented match the analysis plan?

No- The paper didn't demonstrate an analysis plan. 

-Are the results clearly and completely presented?

Despite findings being presented, it's hard to ascertain if they were complete as the study objectives were not stated. Also, it's unclear why urinary schistosomiasis wasn't investigated. It would've been logical to cite previous studies revealing the non-endemicity of urinary schistosomiasis if that was the case in the study area. On the contrary, data from the WHO/ESPEN portal reveals Arba Minch district (the area under investigation) is endemic to urinogenital schistosomiasis.

Kato-Katz and formol ether concentration techniques were both used for stool analysis. It’s unclear if:

- the outcomes were different by technique, 

- one technique was used as first line screening and the other for confirmatory diagnosis. 

It is necessary to state:

- why both techniques were used 

- if prevalence/intensity varied by technique 

- how prevalence and intensity were established using both diagnostic methods. 

-Are the figures (Tables, Images) of sufficient quality for clarity?

Yes- Tables and narratives are clear. However, the results on prevalence and intensity were not disaggregated by gender or school enrolment/attendance. Also, it would've been excellent to appraise the knowledge of survey respondents on disease transmission and prevention. Furthermore, data wasn't disaggregated by the five sub villages in the Shelle Mela Kebele area.

**Conclusions**

-Are the conclusions supported by the data presented?

-Are the limitations of analysis clearly described?

-Do the authors discuss how these data can be helpful to advance our understanding of the topic under study?

-Is public health relevance addressed?

Reviewer #1: Yes, though please provide the recommendation of school-based deworming in the area with a brief justification in the discussion, not just the conclusion

Reviewer #2: Yes.

Reviewer #3: -Are the conclusions supported by the data presented?

Partially- It was concluded that 52.3% of SAC were infected with intestinal schistosomiasis, but the absence of S. haematobium in the study makes it challenging to conclude if S. mansoni is the predominant species in the study area. Also, it would be useful to confirm if STH deworming is required or not, given that Kato-Katz and formol ether techniques were used for stool analysis. 

-Are the limitations of analysis clearly described?

No- Study limitations were not mentioned. 

-Do the authors discuss how these data can be helpful to advance our understanding of the topic under study?

Partially - Study falls short of investigating transmission factors like access to water sanitation and hygiene, including behavioural determinants of transmission amongst participants.

-Is public health relevance addressed?

Partially - Beyond MDA being recommended, the need for health education initiatives was not cited.

**Editorial and Data Presentation Modifications?**

Reviewer #1: As above, please provide prevalence of intensity class using total sampled as denominator as well as using total infected as denominator which is already presented

Reviewer #2: (No Response)

Reviewer #3: Hyperlinks to sample consent forms, IRB requirements and questionnaires could better enhance the appraisal of reviewers.

**Summary and General Comments**

Reviewer #1: Authors have presented an important and well performed survey of S. mansonii (SCH) in a seemingly underserved area of Ethiopia. The alarmingly high prevalence and intensity of SCH within the targeted communities will hopefully provide Government with sufficient evidence to roll out preventative chemotherapy to avoid further morbidity and mortality. The authors are commended on their effort. 

A few minor comments, further to those above:

- Please provide a thorough review language used

- Please ensure references are used where appropriate, for example - methods section states health center registration records were checked. Convention states that a reference should be provided here, even if it is a government data set.

- Methods suggest only households with SAC were included in study. How were households with SAC identified. Was a community register used, then random households selected?

- Could authors provide the timing from sample collection to slide reading? This is important for results relating to hookworm - although identifying hookworm is not the primary objective of the study, would be useful nonetheless

- The calculation for the epg in the methods is repeated (ie mentioning multiplication by 24) this need only be in the article once

- Participant selection suggests samples given only upon consent, please expand a little here. Was informed consent involving information on the study objectives etc given prior to offering participation?

- IRB approval has been obtained for this study, but a statement of the IRB process/approval should be provided in the methods section. 

- Please review the discussion section for some repetition

Reviewer #2: This is a straightforward parasitological study assessing the prevalence of S. mansoni in school children. 

There are no line numbers, making it very difficult to provide detailed comments. 

Title: The second part of the title is not relevant. The area has not received any treatment before. Thus, the high prevalence should automatically indicate the need for a control approach. It is not a question for debate.

There are some typos and grammatical errors that will need to be corrected. Kindly go through the entire text and address them. See the attached for further comments.

The discussion and recommendation could be expanded to consider the treatment of out-of-school children and other community members. Other control options could also be discussed.

The data on the socio-demographic information (other than age and gender) was not presented. The questionnaire should be added as an appendix to the study, and the rest of the data presented.

Please italicize S. mansoni and other scientific names throughout the text.

Reviewer #3: (No Response)

PLOS authors have the option to publish the peer review history of their article (what does this mean?). If published, this will include your full peer review and any attached files.

Reviewer #1: No

Reviewer #2: No

Reviewer #3: Yes: Ndellejong Cosmas Ejong
---

## [Editor Report · Decision Letter 1]

8 Oct 2024

Dear Dr Massebo,

We are pleased to inform you that your manuscript 'High prevalence of intestinal schistosomiasis in school-age children in the villages adjacent to Lake Chamo in the southern Rift Valley of Ethiopia' has been provisionally accepted for publication in PLOS Neglected Tropical Diseases.

Best regards,

Jong-Yil Chai

Section Editor

Jong-Yil Chai

Section Editor

Your revised manuscript is acceptable by PLoS NTD. Thanks for your kind cooperation.

---

## [Editor Report · Acceptance letter]

24 Oct 2024

Dear Dr Massebo,

We are delighted to inform you that your manuscript, "High prevalence of intestinal schistosomiasis in school-age children in the villages adjacent to Lake Chamo in the southern Rift Valley of Ethiopia," has been formally accepted for publication in PLOS Neglected Tropical Diseases.

Best regards,

Shaden Kamhawi

co-Editor-in-Chief

Paul Brindley

co-Editor-in-Chief
